# IDO1/COX2 Expression Is Associated with Poor Prognosis in Colorectal Cancer Liver Oligometastases

**DOI:** 10.3390/jpm13030496

**Published:** 2023-03-09

**Authors:** Miaoqing Wu, Xiaoliang Wu, Xing Wang, Xiangchan Hong, Yifan Liu, Guangzhao Lv, Cong Li, Zhizhong Pan, Rongxin Zhang, Gong Chen

**Affiliations:** 1Department of Colorectal Surgery, Cancer Centre, Sun Yat-sen University, Guangzhou 510060, China; 2State Key Laboratory of Oncology in South China, Collaborative Innovation Center of Cancer Medicine, Guangzhou 510060, China; 3Department of Oncology, Shenzhen Hospital of Southern Medical University, Shenzhen 518000, China; 4First Affiliated Hospital of Jiangxi Medical College, Nanchang 330000, China; 5Collaborative Innovation Center of Cancer Medicine, Guangzhou 510060, China

**Keywords:** colorectal cancer, oligometastases, IDO1, COX2, prognosis

## Abstract

Background: IDO1 and COX2 have emerged as promising immunotherapy targets. It is unclear whether IDO1 and COX2 expression levels in colorectal cancer (CRC) patients with liver oligometastases could be independent predictors of overall survival (OS) and progression-free survival (PFS). The purpose of this study was to investigate the correlation of IDO1 and COX2 expression levels with OS and PFS in CRC patients with liver oligometastases. Methods: The expression levels of IDO1 and COX2 were assessed by immunohistochemistry in 107 specimens from patients with liver oligometastases. The correlation between the expression of IDO1 and COX2 and the clinicopathological parameters and OS/PFS in patients was examined. Results: The expression level of IDO1/COX2 was significantly correlated with age and was not associated with gender, BMI, T stage, N stage, primary tumor size, liver metastasis size, CEA, CA19-9, CD3 TILs or CD8 TILs. In univariate analysis, we found that IDO1/COX2 expression, CEA and N stage all yielded significantly poor OS and PFS outcomes. In our multivariate Cox model, IDO1/COX2 coexpression, CEA and N stage were found to be significantly correlated with OS; IDO1/COX2 coexpression and CEA were significantly correlated with PFS. Conclusions: IDO1/COX2 coexpression plays a pivotal role and may act as a potential prognostic biomarker for survival in CRC patients with liver oligometastases.

## 1. Introduction

Colorectal cancer (CRC) remains a major public health issue worldwide, representing the third most common cancer and second leading cause of death according to 2020 cancer statistics [1,2]. The liver is recognized as the most common site of CRC metastasis [3,4]. Approximately 14–18% of CRC patients are diagnosed at the first medical consultation, and more than 50% of patients with colorectal cancer develop metastasis in the liver [5,6,7,8].

Accumulating evidence suggests that immunotherapy has become one of the most common treatments for CRC [9,10]. A recent study showed that radiofrequency ablation (RFA) increased PD-L1 expression and infiltrating T cells in the tumor microenvironment in patients with synchronous colorectal cancer liver metastases [11]. Through the use of mouse tumor models, the results show that the combined therapy of the RFA and PD-1 blockade synergistically improved T cell-mediated immune responses and tumor rejection [11,12,13]. Several other inhibitory factors suppress T cell-mediated immune responses in the tumor microenvironment. The catabolism of tryptophan is an important pathway for establishing innate and adaptive immune tolerance, which is driven by rate-limiting enzymes, indoleamine-2,3-dioxygenase 1 (IDO1) and tryptophan-2,3-dioxygenase 2 (TDO), resulting in a local decrease in tryptophan and the accumulation of kynurenine and its derivatives [14,15]. T cells in the tumor microenvironment sense low tryptophan and high kynurenine via the mTORC and GCN signaling pathways to initiate an amino acid starvation response, resulting in T cell cycle arrest and favoring the differentiation of regulatory T cells, ultimately forming a profoundly immunosuppressive tumor microenvironment [16].

Cyclooxygenase-2 (COX-2) and prostaglandin E2 (PGE2) expressions drive constitutive expression of IDO1 in many human tumor cells via the PKC and PI3K pathways [17]. The MAPK signaling pathway controls COX2 expression, which indirectly induces IDO1 expression [14]. IFN-r is expressed by activated T cells and can induce IDO1 expression in most tissues and cell types, resulting in an inhibition of T cell responses to tumor cells. Because most tumors carry oncogenic mutations in the MAPK signaling pathway, they may favor constitutive IDO1 expression without IFN-r [17]. Above all, IDO1 and COX2 have attracted attention in cancer research and may be promising prognostic and therapeutic biomarkers of tumor tissues.

IDO1 has been demonstrated as expressing in 58% of human cancers [18], and its expression levels are correlated with the poor clinical outcome in several cancer types, such as melanoma, gynaecological cancers like endometrial carcinoma, liver and ovarian cancers [19,20,21,22,23,24,25]. One recent study showed that cytoplasmic IDO1/COX2 coexpression, but not nuclear IDO1/COX2 coexpression, could be identified as a poor independent predictor for OS in CRC patients [26]. Another study demonstrated that IDO1 inhibition sensitized colorectal cancer to radiation-induced cell death [27]. Furthermore, IDO1 has been confirmed to promote cancer metastasis in gastric cancer and hepatocellular carcinoma [28,29]. However, whether IDO1/COX2 coexpression is correlated with OS and PFS in patients with liver metastases of colorectal cancer remains unknown.

In this study, we conducted a retrospective analysis of the potential prognostic importance of the correlation of IDO1/COX2 expression in OS and PFS in CRC patients with liver oligometastases of colorectal cancer.

## 2. Materials and Methods

### 2.1. Patients and Specimens

This study was approved by the Institutional Review Board and Human Ethics Committee of Sun Yat-Sen University Cancer Center. Written consent for using the samples for research purposes was obtained from all patients before surgery. All tissues were collected from 107 patients who underwent a surgical resection between 1 June 1999 and 1 December 2016 at the Department of Colorectal Surgery of Sun Yat-sen University (Guangzhou, China). Patients from all the groups received colorectal tumor and hepatic oligometastatic resection at different time points. The eligibility criteria were as follows: (1) CRC with hepatic oligometastasis; (2) all tumor tissue pathological diagnoses confirmed to be CRC and liver metastases by a pathologist; and (3) no anticancer therapies received before the operation.

### 2.2. Immunohistochemical Staining

A total of 107 CRC tumor tissues were used in the immunohistochemistry (IHC) analysis. All formalin-fixed, paraffin-embedded tumor specimens were cut into 4 μm sections as previously described [26]. After baking at 60 °C for 2 h, the samples were deparaffinized in xylene and rehydrated in a series of graded ethanol solutions. Then, 3% hydrogen peroxide was used to block endogenous peroxidase activity for 10–15 min. The samples were microwaved for antigen retrieval for 30 min in 0.01 mol/L sodium citrate buffer (pH 6.0), and then preincubated for 30 min in 10% normal goat serum to block nonspecific staining. The sections were incubated with the primary rabbit anti-human COX2 monoclonal antibody (working dilution, 1:200; Beijing Golden Bridge Biotechnology, Beijing, China), rabbit anti-human IDO1 monoclonal antibody (working dilution, 1:100; Cell Signaling Technology, Danvers, MA, USA), mouse anti-human CD8 monoclonal antibody (working dilution, 1:100; Beijing Golden Bridge Biotechnology) and rabbit anti-human IDO1 monoclonal antibody (working dilution: 1:50; Beijing Golden Bridge Biotechnology) overnight at 4 °C. The next day, the samples were incubated at room temperature for 30 min with the secondary antibody (Dako, Glostrup, Denmark).

Assessments of the staining were scored by two experienced independent pathologists blinded to the patients’ identities. H-scores of the percentage of positive tumor cells (0% to 100%) and dominant staining intensity (0 to 3) of immunostaining were used for the data analysis. The final quantitation of each specimen was obtained by multiplying the two scores. COX2 expression was considered high if the score was higher than the median score of 0.6. IDO1 expression was classified as high if the H-score was higher than 0.1. T cell infiltration of tumors was assessed using a semiquantitative estimation of the density of CD8-positive/CD3-positive (CD8+/CD3+) cells. An H-score of 3+/4+ for CD8+/CD3+ expression was considered high [30].

### 2.3. Follow-Up

The last follow-up date was 1 April 2018. All patients (65 males and 42 females) were followed up every 3 months in the first 2 years and every 6 months after that for a total of 5 years. Carbohydrate antigen 199 (CA199), carcinoembryonic antigen (CEA) and abdominal and pelvic ultrasound tests were recommended at baseline and every 3 to 6 months for a total of 2 years, and then every 6 months for a total of 5 years. A colonoscopy is recommended at approximately 1 year after resection.

By the time that follow-up occurred, 74 patients (69.2%) had survived. However, 33 patients (30.8%) died of cancer-related causes at the time of the last follow-up report. Overall survival (OS) was defined as the date of liver resection to the date of death or last follow-up. Progression-free survival (PFS) was measured from the date of surgery until the date of relapse or last follow-up.

### 2.4. Statistical Analysis

The GraphPad Prism (version 7.0; GraphPad Software Inc, La Jolla, CA, USA) and SPSS software packages (version 23.0; IBM Corp, Armonk, NY, United States) were used for statistical analysis. The chi-square test was used to assess the correlation between clinicopathologic characteristics and IDO1 status. Survival curves were assessed using the Kaplan–Meier method, and the log-rank test generated differences between curves. *p*-values of less than 0.05 were considered statistically significant.

## 3. Results

### 3.1. IDO1/COX2 Expression in Patients

To illuminate the biological significance of IDO1/COX2 in patients with liver oligometastases of colorectal cancer, we used immunohistochemical staining to test the expression of IDO1 and COX2 in 107 specimens. We detected high expression levels of IDO1 and COX2 in 72/107 (67.3%) and 69/107 (64.5%) primary CRC specimens, and low levels of IDO1 and COX2 in 35/107 (32.7%) and 38/107 (35.5%), respectively (Figure 1). In 72 IDO1 highly expressed specimens, 55 (76.4%) of them also expressed high level of COX2. Expression of IDO1 and COX2 levels are somehow in a similar pattern in this kind of patient.

### 3.2. Correlation of IDO1/COX2 Expression with Clinicopathological Variables

To gain insights into the role of the IDO1/COX2 expression levels in CRC, we correlated IDO1/COX2 expression levels in 107 patients with liver oligometastases of colorectal cancer using 11 widely recognized clinicopathological features (Table 1). IDO1/COX2 expression level was significantly correlated with age (*p* = 0.004); in contrast, we observed no correlation between IDO1/COX2 expression level and other clinical factors, such as gender, BMI, T stage, N stage, primary tumor size, liver metastases size, CEA, CA19-9, CD3 TILs and CD8 TILs (all *p* > 0.05).

### 3.3. Univariate Analysis of the Correlation of Clinicopathological Parameters with OS

To confirm the effect of traditional clinicopathological parameters on OS in patients with liver oligometastases of colorectal cancer, we performed a univariate analysis of traditional clinicopathological parameters for prognosis. The results reveal that N stage (1/2 vs. 0) (*p* = 0.050) and CEA level (ng/mL) (>5 vs. ≤5) (*p* = 0.024) were significantly correlated with OS in patients with liver oligometastases of colorectal cancer (Figure 2b,c). We observed other clinicopathological parameters, such as gender, age, BMI, T stage, tumor size, liver metastasis size, CA19-9, CD3 TILs, and CD8 TILs, which were not significantly correlated with OS in patients with liver metastases of colorectal cancer (Figure 2d–n). We found that IDO1/COX2 coexpression (Group IV vs. Group I/II/III), as opposed to IDO1 or COX2 coexpressions alone, was significantly correlated with OS (*p* = 0.002) (Figure 2a).

### 3.4. Univariate Analysis of the Correlation of Clinicopathological Parameters with PFS

Kaplan–Meier analysis demonstrated that N stage (1/2 vs. 0) (*p* = 0.026) and CEA level (ng/mL) (>5 vs. ≤5) (*p* = 0.006) were significantly correlated with PFS in patients with liver oligometastases of colorectal cancer (Figure 3b,c). Other traditional clinicopathological parameters, including gender, age, BMI, T stage, tumor size, liver metastasis size, CA19-9, CD3 TILs, and CD8 TILs, were not significantly correlated with PFS in patients with colorectal cancer liver oligometastases (Figure 3d–n). We also performed a univariate analysis to assess whether IDO-1 and COX2 were associated with prognosis. We observed that IDO-1/COX2 coexpression, as opposed to IDO1 or COX2 expressions alone, was significantly correlated with PFS (*p* = 0.0024) (Figure 3a).

### 3.5. Multivariate Analysis of the Correlation of Clinicopathological Parameters with OS

We also performed multivariate Cox modeling to analyze whether traditional clinicopathological parameters and IDO-1/COX2 coexpression represent potential independent predictors for OS outcome in patients with liver oligometastases of colorectal cancer. We observed that CEA level (ng/mL) (>5 vs. ≤5) (*p* = 0.050; HR = 2.137; 95% CI: 1.001–4.566) and IDO-1/COX2 (Group IV vs. Group I/II/III) (*p* = 0.002; HR = 2.315; 95% CI: 1.052–5.102) were significantly correlated with OS for patients with liver oligometastases of colorectal cancer (Table 2). However, IDO-1 or COX2 could not be an individual predictor for OS in multivariate Cox modeling. Other traditional clinicopathological parameters, including gender, age, BMI, T stage, N stage, primary tumor size, liver metastasis size, CA19-9, CD3 TILs, and CD8 TILs, were not significant with OS for patients with liver oligometastases of colorectal cancer (Table 2).

### 3.6. Multivariate Analysis of the Correlation of Clinicopathological Parameters with PFS

We demonstrated that CEA level (ng/mL) (>5 vs. ≤5) (*p* = 0.007; HR = 2.538; 95% CI: 1.291–5.000) and IDO-1/COX2 (Group IV vs. Group I/II/III) (*p* = 0.013; HR = 2.347; 95% CI: 1.197–4.608) were significantly correlated with PFS in patients with liver oligometastases of colorectal cancer (Table 2). However, IDO-1 or COX2 could not be an individual predictor for PFS in multivariate Cox modeling. Other traditional clinicopathological parameters, such as gender, age, BMI, T stage, N stage, tumor size, liver metastasis size, CA19-9, CD3 TILs, and CD8 TILs, were not significant with PFS for patients with liver oligometastases of colorectal cancer (Table 2).

## 4. Discussion

Although current immunotherapy has achieved promising results in many tumor types, such as melanoma and non-small cell lung cancer, more than 50% of cancer patients will progress with resistance to immunotherapy and will need more new therapies. Tumors that avoid immune destruction are dependent on various mechanisms. IDO1 expression is associated with T-cell infiltration and is induced by the IFN-r produced by infiltrating T cells. A recent study also showed that tumor cells that produce IDO1 are constitutively dependent on COX2 and PGE2 via the PKC and PI3K pathways and continuously prevent T-cell infiltration. Therefore, inhibiting IDO1 has become a new target for cancer therapy. One recent study validated that a COX2 inhibitor or an IDO1 inhibitor could reject tumors in NSG mice. In previous studies, high IDO1 expression in CRC was significantly correlated with a reduction in CD3-positive TILs and the presence of metastatic disease, revealing the important role of IDO1 in the therapeutic blockade for this disease. We recently evaluated treated and untreated CRC patients for celecoxib, an inhibitor of COX2, and found that cytoplasmic IDO1 and COX2, but not nuclear IDO1 and COX2, were correlated with OS in patients treated with or without celecoxib. However, clinical studies do not demostrate IDO1 and COX2 coexpression in CRC patients with liver oligometastases of colorectal cancer.

In the present study, univariate analysis, but not multivariate analysis, revealed that N stage was correlated with OS and PFS in CRC patients with liver oligometastases of colorectal cancer. In both the univariate and multivariate analysis results, CEA and IDO-1/COX2 were significantly associated with OS and PFS in our patients. As CRC cells lose their polarity, CEA begins to accumulate on the surface of cells and is then released into the bloodstream. As such, the amount of CEA increases as the tumor size increases. Our results show that CEA might be an independent prognostic factor for CRC patients with liver oligometastases. In our previous study, we found that the coexpression of cytoplasmic IDO-1/COX2 plays a key role in survival prognosis for CRC patients treated with or without celecoxib. Furthermore, in CRC patients with liver oligometastases, IDO-1/COX2 could be an independent prognostic factor for OS and PFS. In this study, individual IDO1 or COX2 had no significant relationship with OS and PFS, and there may be other signaling pathways activating IDO1 expression in CRC patients with liver oligometastases.

In our study, it implied that IDO1 and COX2 could be a feasible marker for prognosis for colorectal cancer patient with liver oligometastases who seemed could benefit from surgery for both primary site and liver metastatic site. Physicians should take further postoperative management into consideration to avoid recurrence incidence.

There are limitations to our study. First, to minimize bias and immunohistochemistry methodological limitations, we adopted rigorous standardized assay methods, and two well-trained and blinded clinical pathologists independently affirmed immunohistochemistry scores. Second, the number of patients is modest. More specimens need to be collected to confirm our results, and more CRC liver oligometastases patients need to be considered. Besides, this study still has its intrinsic associated limitations for its retrospective nature.

## 5. Conclusions

The results of our current study reveal that CEA and IDO-1/COX2 may serve as feasible biomarkers for the prognostic prediction to predict CRC patients with liver oligometastases of colorectal cancer OS and PFS. It hints that active combined therapy and close post-operative surveillance should be considered for the CRC patients with both liver oligometastases and high expression levels of IDO-1/COX2.

## Figures and Tables

**Figure 1 jpm-13-00496-f001:**
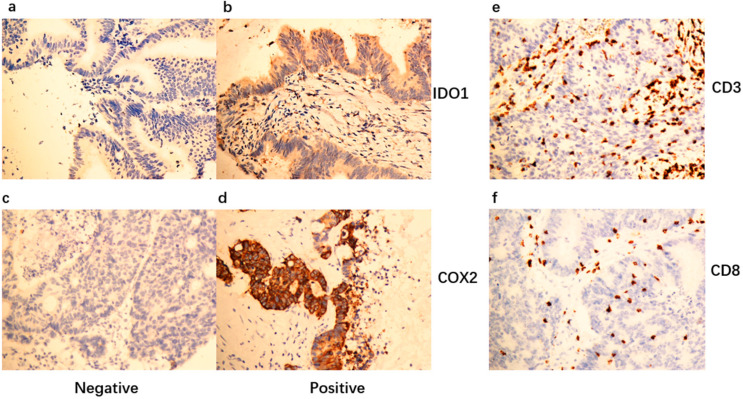
**IDO1, COX2, CD3 and CD8 expression levels in CRC patients with liver oligometastases.** (**a**) Immunohistochemistry (IHC) of negative IDO1 expression in CRC liver oligometastases; (**b**) immunohistochemistry (IHC) of positive IDO1 expression in CRC patients with liver oligometastases; (**c**) immunohistochemistry (IHC) of negative COX2 expression in CRC liver oligometastases; (**d**) immunohistochemistry (IHC) of positive COX2 expression in CRC liver oligometastases; (**e**) immunohistochemistry (IHC) of positive CD3 expression in CRC liver oligometastases; (**f**) immunohistochemistry (IHC) of positive CD8 expression in CRC liver oligometastases.

**Figure 2 jpm-13-00496-f002:**
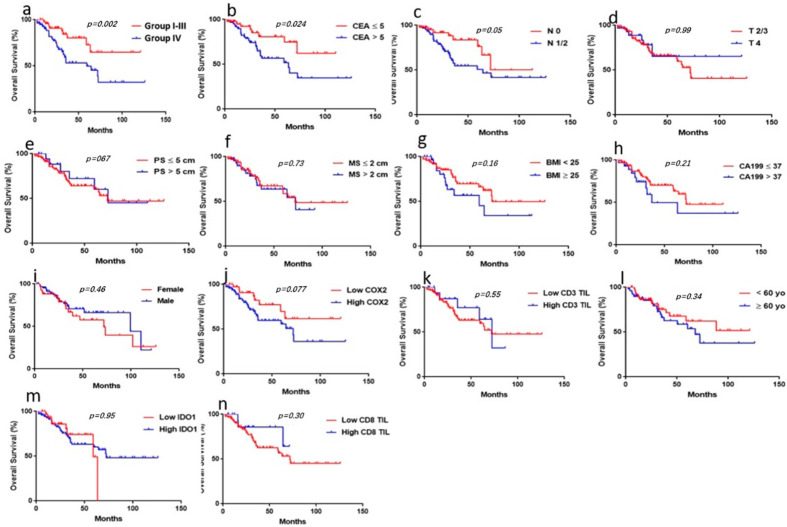
**Correlation of the clinicopathological parameters with OS in CRC patients with liver oligometastases.** Survival curves were generated using the Kaplan–Meier method, and differences between survival curves were estimated using the log-rank test. (**a**) We divided all patients into four groups based on the level of IDO1 and COX2 expression. Group I: IDO1LowCOXLow; Group II: IDO1HighCOXLow; Group III: IDO1LowCOXHigh; Group IV: IDO1HighCOXHigh. The association of the four groups (IV vs. I-III) with OS was significant (*p* = 0.002); (**b**,**c**) correlation between CEA and N stage and OS in patients; b: CEA (*p* = 0.024); c: N stage (*p* = 0.05). (**d**–**n**): Correlation between other clinicopathological parameters and OS in patients. (*p* > 0.05).

**Figure 3 jpm-13-00496-f003:**
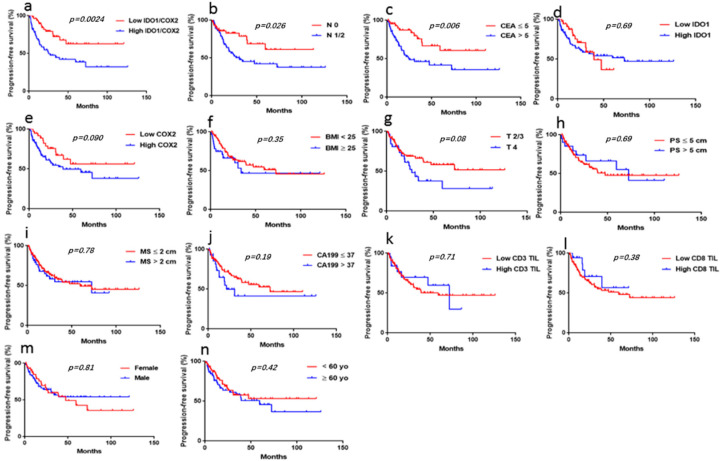
**Correlation of the clinicopathological parameters with PFS in CRC patients with liver oligometastases.** Survival curves were generated using the Kaplan–Meier method, and differences between survival curves were estimated using the log-rank test. (**a**) We divided all patients into four groups based on IDO1 and COX2 expression levels. Group I: IDO1LowCOXLow; Group II: IDO1HighCOXLow; Group III: IDO1LowCOXHigh; Group IV: IDO1HighCOXHigh. The association of the four groups (IV vs. I/II/III) with PFS was significant (*p* = 0.0024); (**b**,**c**) correlation between CEA and N stage and PFS in patients; c: N stage (*p* = 0.026); b: CEA (*p* = 0.006). (**d**–**n**): Correlation between other clinicopathological parameters and PFS in patients. (*p* > 0.05).

**Table 1 jpm-13-00496-t001:** Correlation of IDO-1/COX2 expression with clinicopathological parameters in patients with liver metastases of colorectal cancer.

Characteristics	No. of Patients	IDO-1/COX2 Expression (%)		*p*-Value
		One or Both Low	Both High	
Gender				
Male	65	33 (50.8%)	32 (49.2%)	0.424
Female	42	18 (42.9%)	24 (57.1%)	
Age (years)				
≥60	51	17 (33.3%)	35 (66.7%)	0.004
<60	56	34 (60.7%)	22 (39.3%)	
BMI				
≥25	25	13 (52.0%)	12 (48.0%)	0.620
<25	82	38 (46.3%)	44 (53.7%)	
T stage				
4	32	12 (37.5%)	20 (62.5%)	0.415
2/3	75	39 (52.0%)	46 (48.0%)	
N stage				
1/2	63	28 (44.4%)	35 (55.6%)	0.425
0	44	23 (52.3%)	21 (47.7%)	
Primary tumor size (cm)				
>5	20	9 (45.0%)	11 (55.0%)	0.791
≤5	87	42 (48.3%)	45 (51.7%)	
Liver metastases size (cm)				
>2	44	21 (47.7%)	23 (52.3%)	0.991
≤2	63	30 (46.7%)	33 (53.3%)	
CEA in ng/mL				
>5	62	28 (45.2%)	44 (54.8%)	0.195
≤5	45	23 (51.1%)	22 (48.9%)	
CA19-9 in U/mL				
>37	29	15 (51.7%)	14 (48.3%)	0.608
≤37	78	36 (46.2%)	42 (53.8%)	
CD3 TILs				
High	19	10 (52.3%)	9 (47.7%)	0.633
Low	88	41 (46.6%)	47 (53.4%)	
CD8 TILs				
High	18	10 (55.6%)	8 (44.4%)	0.462
Low	89	41 (46.1%)	48 (53.9%)	

**Table 2 jpm-13-00496-t002:** Univariate and multivariate analysis of the correlation of clinicopathological parameters with prognosis in patients with liver metastases of colorectal cancer.

Variables	OS				PFS			
	Univariate	Multivariate			Univariate	Multivariate		
	*p*-Value	*p*-Value	HR	95% CI	*p*-Value	*p*-Value	HR	95% CI
Gender (Male vs. Female)	NS	NS			NS	NS		
Age, years (≥60 vs. <60)	NS	NS			NS	NS		
BMI (≥25 vs. <25)	NS	NS			NS	NS		
T stage (4 vs. 2/3)	NS	NS			NS	NS		
N stage (1/2 vs. 0)	0.050	NS			0.026	NS		
Primary tumor size (cm) (>5 vs. ≤5)	NS	NS			NS	NS		
Liver metastases size (cm) (>2 vs. ≤2)	NS	NS			NS	NS		
CEA (ng/mL) (>5 vs. ≤5)	0.024	0.050	2.137	1.001–4.566	0.006	0.007	2.538	1.291–5.000
CA19-9 (U/mL) (>37 vs. ≤37)	NS	NS			0.19	NS		
CD3 TILs (High vs. Low)	NS	NS			NS	NS		
CD8 TILs (High vs. Low)	NS	NS			NS	NS		
IDO-1 (High vs. Low)	NS	NS			NS	NS		
COX2 (High vs. Low)	NS	NS			NS	NS		
IDO-1/COX2 (Group IV vs. Groups I/II/III)	0.002	0.037	2.315	1.052–5.102	0.0024	0.013	2.347	1.197–4.608

## Data Availability

All relevant data are within the manuscript and its additional files.

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
