# Peer review of "IDO1/COX2 Expression Is Associated with Poor Prognosis in Colorectal Cancer Liver Oligometastases"

_jpm, 2023, doi:10.3390/jpm13030496_

Round 1

Reviewer 1 Report

The manuscript entitled “IDO1/COX2 Expression Is Associated With Poor Prognosis In Colorectal Cancer Liver Oligo-metastases” of Miaoqing W et al. study the role of IDO1/COX2 as biomarkers in the colorectal cancer prognosis. The manuscript is well organized and of great interest. Unlikely, the conclusion section is poor and I suggest to improve this part for the publication after major revision.

Author Response

Response to Reviewer 1 Comments

Point 1: The manuscript entitled “IDO1/COX2 Expression Is Associated With Poor Prognosis In Colorectal Cancer Liver Oligo-metastases” of Miaoqing W et al. study the role of IDO1/COX2 as biomarkers in the colorectal cancer prognosis. The manuscript is well organized and of great interest. Unlikely, the conclusion section is poor and I suggest to improve this part for the publication after major revision.

Response 1: We agree with the reviewer that further elaborating on this point. Based on the conclusion of our study, IDO1/COX2 could be feasible biomarkers and prognostic predictors for CRC patients with liver oligo-metastases. And we would like to add the possible clinical meaning if these two biomarkers applied to clinical management for the patients. It hints physicians could perform a prudent selection for combined management and close surveillance for IDO1/COX2-high-expressed patients. We have revised the text to address your concerns and hope the conclusion is now clearer. Please refer to page 9 of the revised manuscript, line 281-283.

Reviewer 2 Report

The authors carried out previously a very similar study (Ref.15) to the one shown in the current manuscript on Colorectal cancer but without liver metastasis. They concluded that cytoplasmic IDO1/COX2 coexpression could be a poor independent predictor for OS in CRC.

In this manuscript, they carried out the same retrospective study on Colorectal cancer samples with liver oligo metastasis. They stated the same conclusion.

In the presence of the previous study (Ref 15), the work presented in this manuscript will not add significant knowledge or data to the literature. If this work is to be published, the whole manuscript needs significant English revision and sentence structure. The authors also must consider that most of this work was already published in Ref 15 but on different samples.  

Author Response

Response to Reviewer 2 Comments

Point 1: The authors carried out previously a very similar study (Ref.15) to the one shown in the current manuscript on Colorectal cancer but without liver metastasis. They concluded that cytoplasmic IDO1/COX2 coexpression could be a poor independent predictor for OS in CRC.

In this manuscript, they carried out the same retrospective study on Colorectal cancer samples with liver oligo metastasis. They stated the same conclusion.In the presence of the previous study (Ref 15), the work presented in this manuscript will not add significant knowledge or data to the literature. The authors also must consider that most of this work was already published in Ref 15 but on different samples.

Response 1: In Ref.15(Now Ref.26 in revised version), they had the CRC patients’ sample collected. Patients enrolled in Ref.15 show are all diagnosed as stage II to III colorectal cancer whose prognosis are better than stage IV CRC patients. For our study, the patients enrolled in our study are all stage IV CRC patients, and what’s more specific, with liver oligo-metastasis. For CRC patients, nearly 50% of them will develop liver metastasis during the disease course. Among all the liver metastasized CRC patients, liver oligo-metastasis occupied about 20% according to a population-based study in Sweden. And another study show that surgery is only applicable in 10-20% of liver metastatic cases. Despite liver oligo-metastasized CRC seems been able to received curative surgery for management which could bring survival benefit for this type of patient, in fact, this kind of patients still had 5-year survival rate as low as 30%. It hints that there would be biomarkers correlated to poor prognosis of these types of the patients. Our aim is to find out biomarkers for surgeons who could be able to do a prudent selection for these patients. We believed that, there’s no conflicts for the conclusions between our study and Ref.15(Now Ref.26).

Point 2: If this work is to be published, the whole manuscript needs significant English revision and sentence structure.

Response 2: We tried our best to improve the English editing and made some changes to the manuscript. These changes will not influence the content and framework of the paper. And here we did not list the changes but marked in red in the revised paper.

Round 2

Reviewer 1 Report

The authors improved the manuscript as required. I suggest this work for publication

Reviewer 2 Report

The authors made some changes which added an improvement to the manuscript. The manuscript is now more suitable for publication.